# Modular Lentiviral Vectors for Highly Efficient Transgene Expression in Resting Immune Cells

**DOI:** 10.3390/v13061170

**Published:** 2021-06-18

**Authors:** Christina Fichter, Anupriya Aggarwal, Andrew Kam Ho Wong, Samantha McAllery, Vennila Mathivanan, Bailey Hao, Hugh MacRae, Melissa J. Churchill, Paul R. Gorry, Michael Roche, Lachlan R. Gray, Stuart Turville

**Affiliations:** 1The Kirby Institute for Infection and Immunity in Society, The University of New South Wales, Sydney, NSW 2052, Australia; cfichter@kirby.unsw.edu.au (C.F.); aaggarwal@kirby.unsw.edu.au (A.A.); andrew.wong@emory.edu (A.K.H.W.); smcallery@mcallery.com (S.M.); v.mathivanan@unsw.edu.au (V.M.); bailey.hao@hotmail.com (B.H.); hughmacrae@outlook.com (H.M.); 2STEM College, RMIT University, Melbourne, VIC 3000, Australia; melissa.churchill@rmit.edu.au (M.J.C.); paul.gorry@rmit.edu.au (P.R.G.); 3Department of Infectious Diseases, Peter Doherty Institute for Infection and Immunity, The University of Melbourne, Melbourne, VIC 3000, Australia; Michael.roche@unimelb.edu.au; 4ViiV Healthcare, Abbotsford, VIC 3067, Australia; lachlan.r.gray@viivhealthcare.com

**Keywords:** gene therapy, lentiviral vectors, resting T cells, vpx, MS2-LVLPs, CRISPR-Cas9

## Abstract

Gene/cell therapies are promising strategies for the many presently incurable diseases. A key step in this process is the efficient delivery of genes and gene-editing enzymes to many cell types that may be resistant to lentiviral vector transduction. Herein we describe tuning of a lentiviral gene therapy platform to focus on genetic modifications of resting CD4^+^ T cells. The motivation for this was to find solutions for HIV gene therapy efforts. Through selection of the optimal viral envelope and further modification to its expression, lentiviral fusogenic delivery into resting CD4^+^ T cells exceeded 80%, yet Sterile Alpha Motif and HD domain 1 (SAMHD1) dependent and independent intracellular restriction factors within resting T cells then dominate delivery and integration of lentiviral cargo. Overcoming SAMHD1-imposed restrictions, only observed up to 6-fold increase in transduction, with maximal gene delivery and expression of 35%. To test if the biologically limiting steps of lentiviral delivery are reverse transcription and integration, we re-engineered lentiviral vectors to simply express biologically active mRNA to direct transgene expression in the cytoplasm. In this setting, we observed gene expression in up to 65% of resting CD4^+^ T cells using unconcentrated MS2 lentivirus-like particles (MS2-LVLPs). Taken together, our findings support a gene therapy platform that could be readily used in resting T cell gene editing.

## 1. Introduction

Genetic editing of T cells continues to show great promise in many clinical efforts. The curing of previous recalcitrant cancers using Chimeric Antigen T cells (CAR T cells) is one powerful example of genetic manipulation of T cells [1,2]. Clinical approaches presently use lentiviral vectors for gene editing in this context [3]. Lentiviruses cannot efficiently deliver and integrate genes directly to “resting” T cells, which require significant levels of either “molding/priming” (i.e., activation/stimulation) to be more receptive to genetic manipulation. Alternatively, the delivery of modified lentiviral vectors is proceeded at very high multiplicities of infection (MOI) to overcome many lentiviral limitations. Whilst this enables genetic manipulation of T cells, this does come at a cost, as cell “molding” lowers the future proliferation and engraftment in vivo [4,5] and the use of very high viral MOIs can often render many approaches cost-prohibitive at the clinic. Both can impact treatment at the clinic. For instance, the potency and efficacy of CAR T cells has been shown to rapidly decrease the longer cells are proliferated in in vitro cultures [6]. In addition, inefficiencies of gene delivery combined with lower potency outcomes in vivo collectively decrease the availability of these powerful technologies as they significantly influence cost, i.e., cost in generating significant volumes of lentiviral vectors and cost in the need to generate more T cells per treatment. Rather than molding cells for the reception of older-generation lentiviral vectors or using significantly higher MOIs to saturate existing restriction factors, our aim was to hone lentiviral delivery by “molding” lentiviral particles to the resting T cells.

Herein we demonstrate how fundamental understandings of how viruses interact with their target cell of interest can lead to bioprospecting of key elements to overcome many inefficiencies in lentiviral delivery. In doing so, this shifts the potential of gene delivery or editing to be dependent upon the viral particle. In our efforts in finding pragmatic solutions for gene therapy towards a HIV cure, we present herein an efficient gene delivery and editing platform for resting CD4^+^ T cells. To enable this, we have re-engineered lentiviral particles from the inside-out. On the outside, we have identified conditions where key viral glycoproteins can mediate cytosolic delivery of gene cargo across resting CD4^+^ T cells. Then we leveraged this by re-engineering the inside of a lentiviral particle at two levels. Firstly, we packaged the SIV/HIV-2 protein Vpx within this lentiviral platform to significantly increase integrative gene delivery into resting CD4+ T cells by antagonizing the lentiviral restriction factor SAMHD1. Secondly, in recognition of the power of gene editing tools like CRISPR-Cas9, we have re-engineered the lentiviral particle for transient delivery of RNA into the cytosol of resting CD4^+^ T cells. In all cases, significant integrative and transient gene delivery was observed. Whilst this can indeed be developed for a portfolio of gene editing tools for HIV cure efforts, it also represents a strategy that can be applied to resting immune cells in general. The key to the latter will be the further bioprospecting of viral elements for incorporation into lentiviral vectors.

## 2. Materials and Methods

### 2.1. Generation of Plasmids for in Cis Vectors and Their Subsequent Production

In cis vectors refer to particles generated following plasmid stratification, whereby structural and enzymatic genes associated with the vector, including an envelope glycoprotein, and Rev, are integrated within a single plasmid. In the context of viral vectors pseudotyped with lentiviral glycoproteins, a plasmid conducive of in cis vector production was generated following the directional subcloning of *env* genes into pCMV-ΔR8.91 (a gift from Didier Trono) using BamHI and EcoRI restriction digestion. Plasmids were modified as previously described to accommodate Vpx into newly formed vectors [7]. Briefly, *gag* was modified proximal to the 3′ p6 region to encode a Vpx recognition motif, 5′-DPAVDLLKNY-3′, adapted from SIVMac239. For the transient expression of Vpx, the SIVmac239 variant of Vpx was subcloned onto pcDNA5.1 by directional cloning utilizing BamHI and XhoI restriction enzymes, generating pcDNA5.1-SIVmac239Vpx. Similarly, for the generation and incorporation of beta-lactamase (BLam), we utilized pMM310 (a gift from Michael Miller, AIDSReagent program). All restriction digests were performed using New England Biolabs (NEB) restriction enzymes.

To produce vectors, we utilized transient transfection of HEK293T cells using polyethylenimine (PEI; PEI MAX 40,000 MW, Polysciences, Warrington, PA, USA). Per transfection reaction, 10.5 μL of 1 mg/mL PEI were added to 1 μg of total DNA diluted in 0.9% NaCl to achieve a final reaction volume of 150 μL. PEI–DNA complexes were allowed to form for 30 min at room temperature. Subsequently, 1 × 10^6^ HEK293T cells were added to the DNA–PEI solution and incubated for another 5 min. The DNA–PEI-cell solution was transferred into a well of a six-well plate, containing 2 mL of pre-warmed DMEM supplemented with 5% heat-inactivated FBS. HEK293T cells were maintained at 37 °C in a humidified atmosphere of 5% CO_2_. Sixteen hours post-transfection, the supernatant was discarded and replaced by fresh DMEM supplemented with 5% FBS. Supernatants were harvested 48 h post-media exchange, and cellular debris were clarified from supernatants by brief centrifugation (2000× *g* 4 °C 5 min). Supernatants were then stored at −80 °C before use. Control vectors were produced by substituting plasmids with salmon sperm DNA or, where appropriate, the non-functional Vpx A3 and its restoration-of-function variant, Vpx A3R [8].

For counting lentivirus particles, wells of a 96-well SensoPlate, glass bottom, black (Greiner Bio-One International GmbH, Kremsmuenster, Austria), were coated with poly-L-lysine according to manufacturer’s instructions (Sigma-Aldrich, St. Louis, MO, USA). Subsequently, 50 µL of virus supernatant was serially diluted at 1 in 5 dilutions steps and then added to each well. The plate was spun at 2500× *g* for 40 min at room temperature followed by fixation with 4% formaldehyde (*v/v*) for 20 min at room temperature. Detection of virus particles were then carried out as previously described [9] with the exception that particles were directly immunostained using KC57-FITC at a final dilution of 1 in 20. Particles were then imaged using DeltaVision Elite microscope fitted with an Evolve 512 EMCCD and detected using an Olympus 60× oil objective NA 1.42. A total of five fields of view per well were acquired in the FITC channel. Viral particles were then enumerated using ImageJ using the 2D/3D particle tracker in MosaicSuite (MOSAIC Group, Dresden, Germany). Unless otherwise stated, for all transductions, we used approximately 6.7 × 10^7^ particles per 1 × 10^6^ resting CD4^+^ T cells, and in terms of p24 concentration, this is calculated to be approximately 100 ng/mL under the assumption that there are 2000 copies of p24 per virion.

### 2.2. Flow Cytometric Characterization of Intracellular SAMHD1 Levels and Percent Transduction

1 × 10^6^ resting CD4 T cells purified from whole blood were plated in U-bottom 96-well plates at a density of 10^6^ cells/mL. Following challenge with control preparations or with vectors containing Vpx, SAMHD1 levels were determined by flow cytometry as previously described [10]. Briefly, cells were first fixed with 1.2% PFA, before permeabilization (Perm Buffer III, BD, Franklin Lakes, NJ, USA) at 4 °C, in the dark for 2 min.

Following two successive cell washes, cells were incubated in the dark with 125 ng mouse anti-SAMHD1 antibody (EMD Millipore MABF933) at 4 °C for 30 min. Cells were then washed twice with ice-cold PBS to remove unbound antibodies, and cells were counterstained with a goat anti-mouse IgG conjugated to AlexaFluor 647 (Thermo Fisher Scientific, Waltham, MA, USA; A32728). Cells were then stored in 2% PFA until analysis. Transduced cells were briefly spun in a centrifuge and pelleted. Media were then aspirated, and cell pellets were washed twice in PBS, then stored in 2% PFA until analysis. Flow cytometry acquisitions were made using a BD LSR II flow cytometer.

### 2.3. Design of HIV-1-Derived MS2-LVLPs and Sensor RNA Construct

The HIV-1-derived packaging vector pNL4.3DY-NC-MCP was generated by substituting the second zinc finger (ZF) domain of nucleocapsid (NC) with the MCP, a concept first described by Prel et al. [11]. Therefore, the SpeI-SbfI fragment of pNL4.3DY was subcloned into pLitmus29 and a gBlock encoding the MCP flanked by AvrII and BglII restriction sites was designed to replace the AvrII-BglII fragment located within the NC. After subcloning into pLitmus, the plasmid was SpeI/SbfI digested and reassembled into pNL4.3DY to generate the final first-generation packaging construct pNL4.3DY-NC-MCP. In order to produce second-generation MS2-LVLPs, a CMV driven packaging construct was generated by cloning the MCP-containing SpHI/SbfI fragment of pNL4.3DY-NC-MCP into psPax2. Highly fusogenic HIV-1 envelope glycoprotein ORFs were subsequently cloned into psPax2-MCP by using SbfI/XhoI restriction sites to generate psPax2-MCP-F39, psPax2-MCP-NL43, psPax2-MCP-Spln3, psPax2-MCP-spln7, and psPax2-MCP-spln12. The resulting size of the final vectors is 13 kb. An RNA sensor construct (pMS2-12X-SL-BlaM) was engineered by cloning 12 repeats of MS2 stem-loops from the pSL-MS2-12X (Addgene #27119) construct into pMM310 via BamHI/XhoI restriction digest.

### 2.4. Production of MS2-LVLPs

First-generation and second-generation MS2-LVLPs were produced following similar methods as described earlier. Specifically, to produce first-generation MS2-LVLPs, 800 ng of packaging-incompetent pNL4.3DY-NC-MCP were co-transfected with 200 ng pMS2-12X-SL-BlaM, encoding 12 MS2 stem-loops fused to beta-lactamase. Packaging controls that lack the NC-MCP fusion (pNL4.3DY) and the Y-deletion (pNL4.3) were generated correspondingly. Positive controls were generated by co-transfecting 800 ng pNL4.3 with 200 ng pMM310 (Addgene #80053), encoding the BLam-vpr fusion protein. Second-generation MS2-LVLPs were produced by co-transfecting 800 ng of CMV-driven psPax2-MCP_envs and 200 ng of pMS2-12X-SL-BlaM.

### 2.5. Isolation of Resting CD4^+^ T Cells

Human Peripheral Blood Mononuclear cells (PBMCs) were purified from whole blood obtained from healthy donors. Blood was immediately diluted 1:2 in Dulbecco’s phosphate buffered saline (DPBS), and PBMCs were isolated using Ficoll-Paque Plus reagent. A commercial CD4 T cell isolation kit (Miltenyi Biotec, Bergisch Gladbach, Germany) was subsequently used to purify resting CD4 T cells from PBMCs by means of negative selection. In brief, non-CD4 T cells were firstly labelled using a cocktail of biotin-conjugated antibodies and secondly with magnetic MicroBeads, enabling isolation of highly pure, untouched CD4 T cells. Cells were utilized within 24 h of isolation.

### 2.6. Fusion Assay

The amount of cell entry was determined by a fluorometric assay, as previously described [12]. Briefly, unconcentrated lentiviral supernatants were spinoculated onto primary resting CD4^+^ T cells purified from whole blood at 1200× *g* and 18 °C for 1 h. Following spinoculation, plates were transferred to an incubator for two hours to allow for VLP/MS2-LVLP entry and release of beta-lactamase protein/mRNA payload into cytoplasmic space. Next, cells were washed and incubated with CCF2-AM (Thermo Fisher Scientific, Waltham, MA, USA, K1025), a beta-lactamase-cleavable compound, that can report cellular fusion with VLPs. The change in emission fluorescence of CCF2 after cleavage by BLam-Vpr chimera or RNA-derived BLam was monitored by flow cytometry. As a control, AMD3100, a CXCR4 inhibitor, was utilized at 10 µM to block gp120-CXCR4 interactions.

### 2.7. Statistical Analysis

All statistical analyses were completed in GraphPad Prism Version 8.2.1 (GraphPad software, San Diego, CA, USA). Data from two groups that were not normally distributed were analyzed using a two-tailed Mann Whitney test whereas data from multiple groups were analyzed using an unmatched/unpaired one-way ANOVA with Dunnett’s post-test. All data are displayed as mean with SEM. *p*-values <0.05 were considered significant.

## 3. Results

### 3.1. Rational Vector Design Platforms to Increase Transduction Potential in Resting CD4^+^ T Cells

Presiding methods largely focus on the transduction of activated T cells [13,14], and thus develop a product that may be suboptimal in function due to prior activation and proliferation that may lead to cellular exhaustion and anergy. This frames a limited window of therapeutic effect due to limitations to those cells’ ability to expand and/or function in vivo (Figure 1A). The utility of resting T cells in the gene/cell therapy arena thus presents several clear advantages. However, there are several factors that impose firm limits on resting T cell therapy being widespread and can be divided into two distinct spatial-temporal events. Firstly, current vectors utilize pseudotypes that are biologically incompatible with attaining entry in resting cells at two levels, initially due to low receptor density on the cellular surface [15]. Secondly, there is limited membrane turnover and particle passage into resting T cells [16]. Despite vector binding, this could provide a significant bottleneck for glycoproteins that require specific vesicular triggers (e.g., lowered pH or cathepsin cleavage). The second major limitation of gene delivery in resting T cells is beyond the membrane. This presently consists of known and unknown number of lentiviral restriction factors that influence vector core uncoating, reverse transcription, integration, and finally gene expression. Thus, any attempt to efficiently genetically manipulate resting T cells needs to make the vector particle compatible to these two key roadblocks (Figure 1B). In addition, this philosophy can be expanded to gene therapy vectors in general: No one vector particle is able to efficiently meet the needs of all cell types, and rather than retrospectively adjusting “the playing field” by changing the cell (e.g., T cell activation or electroporation), we need to consider the foundations of the vector particle itself. As to how we can change these foundations, we need to look closely at how viral families already navigate these challenges and to what extent we can safely bioprospect and harvest key elements to build a vector particle that is better suited for its job of gene delivery and/or editing.

In addressing the at-membrane restriction, we considered pseudotypes that were endocytosis-independent, and were not contingent on membrane fluidity to mediate entry (Figure 1C). Curiously, glycoproteins that are known to target T cells in vivo dominated. Previously we have observed HIV-derived envelopes to efficiently fuse with the membrane of resting CD4 T cells [16]. Therefore, we utilized our extensive back catalogue of phenotypically characterized HIV envelopes (>500 envelopes) to ‘prospect’ for envelope proteins that displayed characteristics desirable to enhance gene delivery in resting T cells. These characteristics included greater affinity of envelopes for the CD4 receptor [17]; enhanced fusogenicity [18]; reduced dependence on CD4 [19,20]; reduced dependence on CCR5 [21]; ability to mediate efficient infection in CD4^+^ T cells [22]; and enhanced tropism for more quiescent CD4^+^ T cell subsets [23,24,25]. To resolve this further, we screened the results from a library of phenotypically characterized HIV-1 glycoproteins that were screened using the HIV Affinofile assay [26,27], which provides quantitative metrics of the ability of given envelopes to scavenge low levels of CD4 and CCR5 for viral entry. This then enabled a shortlist of potential lead candidates (Table 1) based on the selection of glycoproteins that should mediate entry despite low CD4 concentrations, as this was a surrogate for CD4 affinity and high fusogenic activity in resting CD4^+^ T cells.

We next utilized these glycoproteins to generate viral particles, and subsequently screened these preparations on primary resting immune cells, with the hope to identify a lead glycoprotein variant with reliably and consistently high fusogenic potential. Ultimately, this lead glycoprotein would be used as a foundation to enable the successful targeting of resting cells regardless of the intended final application.

It became immediately clear that whilst these shortlisted pseudotypes were substantially more capable of entering resting T cells compared to incumbent pseudotypes, the frequency of entry was still too low for practical use. We realized suboptimal entry may be the consequence of present vector production methods. Specifically, during vector production, all genetic components of vector are separated across several plasmids. During transfection, this may result in ‘plasmid mosaicking’, whereby cells are not all equally transfected with the same quantity of each plasmid, if not all plasmids required. In addition, viruses have evolved to assemble structural proteins and envelope glycoproteins where the expression of each is timed in a similar manner as it is regulated by the same promoter. We thus turned to a simplified two-plasmid system, with three out of the four components stratified onto a single plasmid. We termed this latter strategy in cis given that the gene cassette encoding viral envelope and *tat* were coalesced into the same plasmid encoding the other lentiviral structural and enzymatic genes (Figure 1D). We observed an average three-fold higher transduction outcome when utilizing vector preparations from cells transfected in cis (Figure 1D). As a whole, this rational platform paved the road for committing gene delivery and genetic modification in resting cells, as the vectors generated from this process overcame the limitation of membrane restriction.

### 3.2. Augmentation of Vectors to Increase Transduction Efficacy in Resting CD4^+^ T Cells

Further suppression of vector function is mediated by intracellular restriction factors. One key restriction factor, SAMHD1, has been determined to be a major contributor to resistance against lentiviral transduction, particularly in terminally differentiated myeloid cells such as macrophages, dendritic cells, and also resting T cells [30,31] (Figure 2A). SAMHD1 exists within the nucleus and in the cytoplasm, as dimeric and multimeric forms, which have been shown to degrade nascent transcripts, lower dNTP levels to concentrations that inhibit reverse transcription and integration, and possess innate immune functions, such as the ability to sense and bind to intracellular single-stranded nucleic acids [32,33]. In resting cells, SAMHD1 is present in an active, unphosphorylated form, whilst in dividing cells, SAMHD1 is phosphorylated and largely inactive. Thus, the inactivation or the elimination of unphosphorylated SAMHD1 may well increase the transduction opportunity of each vector. Many SIV and related HIV-2 isolates have evolved the ability to counter SAMHD1 restriction. In brief, an ancestral lineage of SIV originally duplicated the Vpr accessory gene and from that duplication event Vpx evolved and is present in modern lineages of SIV and HIV-2 [34]. How this could be harvested and used in HIV-1-based lentiviral constructs was observed when Sunseri and colleagues identified the minimal motif located in the SIV Gag p6 that could enable Vpx binding and packaging in viral particles [35].

Through incorporation of this Vpx binding motif in HIV-1-based Gag and expression of Vpx in trans, SAMHD1 could be readily countered using existing lentiviral platforms (Figure 2B) [7]. The expression of Vpx and its subsequent packaging into vectors allows for the transient delivery of only the Vpx protein into cytoplasm of target cells following vector entry. Delivery of Vpx into resting CD4^+^ T cells through the use of lentiviral particles carrying the protein Vpx derived from SIVmac239 results in the removal of a majority of the SAMHD1 pool from both nuclear and cytoplasmic compartments, as determined by flow cytometry three days post-Vpx delivery (Figure 2C). Whilst removing the majority of SAMHD1 could increase lentiviral gene delivery anywhere between up to 6-fold compared to non-Vpx lentiviral controls in resting CD4^+^ T cells (Figure 2D), maximal transduction rates were often less than one third of total cells challenged. In contrast, we have observed transduction rates in excess of 80% in both macrophages and dendritic cells [7]. Suboptimal transduction of resting CD4^+^ T cells was not related to incomplete SAMHD1 depletion, given the sustained level of high SAMHD1 depletion within these cells (Figure 2E). To conclude, it was apparent that there was a cell-type difference in post-entry lentiviral restrictions. In myeloid cells, this could be primarily overcome by the delivery of Vpx and subsequent depletion of SAMHD1. In resting T cells, only a proportion of cells were permissive following SAMHD1 depletion. Given this observation, this may point to the timing and/or mechanism of SAMHD1 lentiviral restriction to differ between myeloid and lymphoid cells. To test this hypothesis, we developed viral particles for the delivery of Vpx at and around the time of lentiviral delivery in resting T cells. If timing was a factor influencing resting CD4 T cell transduction, then pre-depletion of SAMHD1 may significantly lentiviral influence transduction rates. Whilst Vpx delivery pre-depleted SAMHD1 in resting CD4^+^ T cells, this did not increase the overall transduction rates to that observed in macrophages or dendritic cells (Figure 2F). In this context, we thus hypothesized that existing restriction factors exist in resting CD4 T cells that are independent to SAMHD1.

The observations of SAMHD1-dependent and -independent restrictions with resting T cells is primarily reflective of vectors that need to complete full reverse transcription and then integrate their cargo. Whilst our observations support a present limitation in integrative delivery of genes, they also point to the observation of highly efficient cytosolic delivery into resting T cells. This latter observation may indeed support the delivery of other gene editing platforms that do not require gene integration like that observed in existing integrative lentiviral vectors. Indeed, many transient platforms have emerged that can deliver CRISPR-Cas9 components for either gene disruption or base gene editing [36,37]. The importance of these latter emerging technologies is their ability to specifically edit genes of interest with limited to no off-target effects.

### 3.3. Engineering of a Non-Integrative, LVLP-Based RNA Delivery Platform to Circumvent Dominant Lentiviral Restrictions Factors in Resting CD4^+^ T Cells

To determine the potential of expressing transgenes without formally necessitating reverse transcription, we engineered non-integrative MS2 lentivirus-like particles (MS2-LVLPs) by fusing the MS2 coat protein (MCP) to the nucleocapsid (NC), as shown in Figure 3A. Importantly, these first-generation MS2-LVLPs are lacking the ability to package genomic RNA due to the absence of the HIV-1 packaging signal Y. Fusion of the MCP monomer (iGag-MCP) or a head-to-tail dimer (iGag-MCP(2×)) to the C terminus of p17 did not result in efficient particle production 72 h post-transfection (data not shown). To produce MS2-LVLPs that efficiently package sensor RNA, we fused 12 MS2 stem-loops to beta-lactamase on a CMV-driven RNA transfer vector (pMS2-12X-SL-BlaM) and co-transfected the construct alongside the first-generation LVLP plasmid (pNL4.3DY-MCP-NC) into HEK293T cells. As schematically illustrated in Figure 3B, the delivery of the vpr-BlaM fusion protein packaged by pNL4.3 served as a positive control to benchmark the RNA delivery efficiency of MS2-LVLPs against the delivery efficiency of BlaM protein. The transduction of resting CD4^+^ T cells with unconcentrated first-generation MS2-LVLPs resulted in a delivery efficiency of biologically active sensor RNA of 29.45%, whereas the efficiency of NL4.3DY and NL4.3 to package sensor RNA was significantly reduced (Figure 3C,D). To investigate whether the transient expression of sensor RNA is solely dependent on a gp120-CXCR4-based entry mechanism, we incubated resting CD4^+^ T cells with the CXCR4 antagonist AMD3100 prior to transduction, which resulted in a reduction of delivery efficiency to <1% (*p* < 0.0001).

### 3.4. Highly Fusogenic Clade C HIV-1 Envelope 1109_F39 Enhances Cell-Specific RNA Delivery to Resting CD4^+^ T Cells

Once the successful delivery of sensor RNA to resting CD4 T cells using first-generation MS2-LVLPs was verified, we moved the MCP-Gag polyprotein gene cassette into the earlier described in cis construct, resulting in a second-generation MS2-LVLP with improved safety features and enhanced fusogenic activity. The safety advantages of MS2-LVLPs are two-fold; firstly, these non-integrative delivery vectors eliminate the imminent risk associated with insertional mutagenesis as the delivery of RNA does not involve a DNA intermediate and thus excludes the possibility of DNA integration events, and secondly, transient expression of transgenes allows for a dose-controlled delivery of functional RNAs into target cells. The latter plays an important role when modifying the system to a DNA editing platform in which CRISPR/Cas9 components are delivered as RNA, hence requiring precise control of intracellular Cas9 activity over dosage and time. To improve the fusogenic activity of second-generation MS2-LVLPs, we cloned five lead envelope glycoprotein ORFs into the in cis Gag.Pol.Tat plasmid. The Gag.Pol.Tat.Env plasmids were co-transfected into HEK293T cells alongside pMS2-12X-SL-BlaM to generate spln3-MS2-LVLPs, spln7-MS2-LVLPs, spln12-MS2-LVLPs, F39-MS2-LVLPs, and NL4.3-MS2-LVLPs. The unconcentrated supernatant was subsequently used to transduce freshly isolated resting CD4^+^ T cells, and transduction efficiencies were determined via the fusion assay. An increase in sensor RNA delivery was notable for all pseudotype candidates, with F39-MS2-LVLPs leading to transient expression of sensor RNA in over 65% of primary resting CD4^+^ T cells (Figure 3E,F). This observation and those outlined above provide several important proofs of principle. Firstly, membrane restrictions in resting T cells can be readily overcome by re-engineering the outer components of lentiviral particles by using T cell tropic envelopes. Secondly, re-engineering of the inner components of the lentiviral particle can enable RNA delivery and transient expression far beyond that observed with integrative lentiviral vectors. The latter importantly highlights the lack of cellular restrictions in transient RNA delivery and protein expression, and moving forward, this platform would be ideal to use with many evolving gene editing methods.

## 4. Discussion

Presently, the gene/cell therapy space is evolving from a niche and exploratory field, into a mature one that promises next-generation personalized healthcare, promising cures or treatments to diseases previously thought to be incurable or untreatable [38,39]. Inroads into the successful treatment of cancers [40,41], blood, and other congenital disorders [42] by utilizing gene-delivering vectors are exemplary of the potential of gene editing technologies to increase quality of life and life expectancy. Gene/cell therapy is fast becoming a new modality in medicine [43]. The heart of gene/cell therapy is the gene-delivering vector, which functions as the sole means of producing a healthcare effect. Today, there are many different gene-delivering vectors, each with their own unique advantages and applications. Thus, there is often the burden of choice during the development of gene/cell therapies as to which vector is best to mediate a therapeutic effect. Lentivirus-mediated gene therapy has shown promising results in recent years [44,45,46] and has been established to be one of the preferred tools for ex vivo gene delivery, showing wide adoption to treat human diseases. Therefore, we strongly argue in support of leveraging lentiviral vectors to mediate gene/cell therapy. Nonetheless, we are also cognizant that there are limitations to these vectors. Lentiviral vectors mediate permanent genetic change. As such, their propensity and therefore suitability for inserting genes into cells, especially hematopoietic stem cells, make lentiviral vectors an obvious tool to facilitate long-lived therapy. Yet, lentiviral vectors have experienced setbacks in recent years. There is a high likelihood of patients experiencing high-grade adverse events during therapy [47], and there is the persisting risk of oncogenic transformation [48]. Concerns over safety notwithstanding, and as we have shown, lentiviral vectors often have difficulties transducing certain cell types, which limits the range of diseases that can be successfully treated using gene/cell therapy. The requirement for cellular activation to create favorable intracellular environments conducive to transduction may come at the cost of cell phenotype modification and reduced long-term effectiveness. Present treatment methods are also protracted and necessitate dedicated infrastructure and highly trained medical personnel to facilitate treatments. There is a push to develop a better understanding of the safety profile of lentiviral vectors and engineer effective fail-safes. Yet, it is still clear that lentiviral vectors are far from perfect. As we have shown, the potential to leverage lentiviral vectors as mediators of transient gene expression and/or couriers of pre-synthesized, folded, and functional proteins provides immense value to the continued use of these vectors in the gene/cell therapy space. Given that gene integration is no longer necessary, risks of oncogenic transformation are also quelled, since there are no components within the vector that can drive oncogene expression (Figure 4A).

To improve the efficiency of current vector production, we have demonstrated that coalescing multiple elements within one construct (in cis) significantly increases transduction efficiencies. Whilst this system represents in vitro proof of principle, future fragmenting of elements may be required to increase safety for clinical use. A similar two-plasmid in cis system, in which the vesicular stomatitis virus protein G (VSV-G) envelope is incorporated into the packaging plasmid, has been described by Levine and colleagues [49]. To increase the biosafety of the two-plasmid system, the authors included a hairpin ribozyme derived from the tobacco mosaic ringspot virus to decrease opportunities for recombination. Such a system could serve as a model for future two-plasmid in cis constructs. Furthermore, we support a plasmid that incorporates Gag and Env on the same plasmid with elements of *pol* expressed as a Gag–Pol fusion protein. Given the latter Gag–Pol fusion protein is expressed at a ratio of 1 copy per 20 Gag as a consequence of a 3′ ribosomal shift site in Gag [50], expression from the Gag–Pol fusion cassette would not need to be high.

Pseudotyping remains one of the key lentiviral vector properties. Thus, vectors can be produced that are targeted to a narrow range of cell types using existing viral envelopes, or perhaps targeted to cells that express certain markers, through the use of custom-built pseudotypes [51]. Such a premise would hold a resounding advantage over other gene/cell therapy vector modalities. By utilizing vectors to induce transient gene expression or to deliver prefabricated proteins to a limited subset of cells, off-target effects could also be minimized if not avoided completely. This raises the possibility of utilizing vectors in vivo, which would dramatically decrease treatment complexity and minimize any donor-to-donor variation that arises from incumbent cell manufacturing methods. In effect, this notion would turn lentiviral vectors into a material that is akin to inert transfection agents, with the added benefit of being able to target very specific cell types. An in vivo gene delivery approach using pseudotyped lentiviral vectors or MS2-LVLPs, however, could face a number of hurdles, including pre-existing immunities through the existence of antibodies in HIV patients as well as the potential of vector components to possess immunogenic properties. Despite the therapeutic potential, these challenges remain to be addressed and will require further vector engineering before an in vivo platform can be considered.

The benefits of transient gene delivery do not signal a complete departure from integrative gene therapy. If the need arises to express therapeutic genes long-term, non-integrative gene therapy can rapidly create a test bed, establishing a best-case scenario to determine the efficacy of the intended therapeutic gene in marriage to a pseudotype. The multiple shared attributes between both modalities are beneficial in the long-term, as it creates a common platform for which vectors can be designed and quickly deployed. Thus, this new modality would be of the greatest utility in the treatment of diseases that would not warrant repeated dosing, such as cancers or viral hepatitis, and embrace a ‘one-and-done’ approach to treatment. Such an approach is also highly relevant to processes involving CRISPR-Cas9 (Figure 4B). Genetic editing through the CRISPR/Cas9 pathway would not necessitate continuous expression of Cas9 enzyme and RNA guides, since the genetic modifications are committed over a single round of treatment. In addition, CRISPR/Cas9 has significantly evolved as a technology, and this can be seen well beyond simply a gene disruption tool to generate gene knock-outs. Firstly, co-delivery of homologous DNA template at the time of CRISPR/Cas9 scission can lead to the replacement of existing genes or the correction of those that may be associated with a genetic disorder. This has recently been used in genetically engineered T cells, where replacement of the TCR with a CAR construct has significantly increased the potency of the engineered T cells in cancer clearance [52]. Whilst the transient system developed and described herein for restricting T cells could provide the CRISPR/Cas9 element, a homologous DNA template could be provided by either adenoviral vectors or integration-deficient lentiviral vectors where DNA would accumulate in the nucleus as 2-LTR circles.

## 5. Conclusions

In short, we describe a gene therapy platform built upon the well-studied and characterized lentiviral vector. While the intention for this platform was to be modular, as to capture a wide gamut of gene/cell therapy needs, we firstly focused on developing a vector with high affinity for resting CD4^+^ T cells. Through improvement in the vector synthesis process, our vectors could mediate cellular entry comfortably in excess of 80% of the targeted pool. Faced with the intracellular restriction factor SAMHD1, we could couple protein cargo within our vectors to overcome SAMHD1-imposed restriction and observed up to 6-fold increase in transduction, with a zenith of over 35% transduction. With the understanding that in order to achieve even higher rates of transgene expression it would be necessary to bypass the biologically limiting steps of reverse transcription and integration, we adapted our particles to coalesce mRNA to direct transgene expression in the cytoplasm shortly after vectors attain cellular entry. In this setting, we observed expression of biologically active mRNA in up to 65% of resting CD4^+^ T cells. To our knowledge, this is the first demonstration of a highly efficient LVLP-based RNA delivery platform for resting CD4^+^ T cells. With our initial goal of opening up the tools needed to modify resting CD4^+^ T cells, we have reached a benchmark that can now be used for the delivery of gene editing tools, and future experiments will be performed to explore this potential. In the context of efforts towards a HIV cure, this may indeed be compatible with initial disruption of latent proviral DNA. If we then couple this with the expression of an HIV resistance motif (simian TRIM5-Alpha gene) or cellular modification event (CCR5 disruption), we could start to influence the HIV reservoir at many levels. Importantly, as the cells are resting and have high proliferative potentials, they could indeed be re-infused to re-seed the in vivo CD4 compartment.

## Figures and Tables

**Figure 1 viruses-13-01170-f001:**
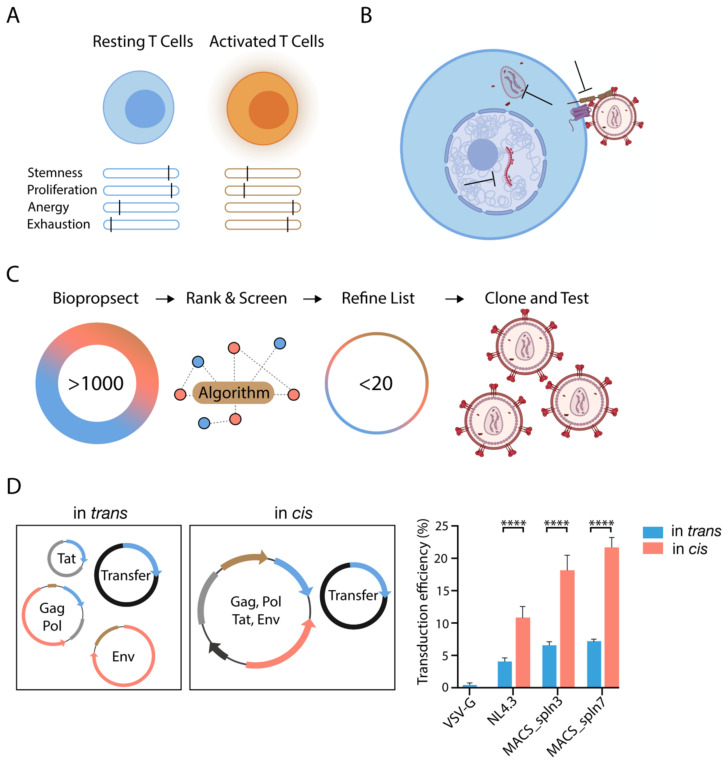
Creation of a lentiviral vector pseudotype discovery pipeline to overcome lentiviral restriction at the plasma membrane imposed by resting CD4^+^ T cells. (**A**) Activation results in T cells that have considerably reduced proliferative potential post-infusion, and increased anergic/exhaustive states that limits effector function. (**B**) Lentiviral vectors are identified to be restricted in resting CD4^+^ T cells at three levels: Firstly, at the membrane through limiting vector entry, secondly within the cytoplasm through limiting reverse transcription and sensing of nascent transcripts, and thirdly within the nucleus by limiting integration. (**C**) A selective pseudotype identification strategy was implemented to determine a pseudotype with the highest fusogenic potential in resting CD4^+^ T cells. Pseudotypes were initially ranked algorithmically, then the highest ranked pseudotypes were tested experimentally for their fusion potential, resulting in the selection of a limited pool of highly fusogenic pseudotypes for further utilization. (**D**) Representation of in cis and in trans vector production methods. In cis expression results in a significant increase in end-point transduction compared to in trans expression, tested orthogonally in three pseudotyping approaches. VSV-G pseudotyped lentiviral vectors were included as negative control. Averages of three patients, with reactions performed in duplicate, are shown. Statistical tests were Student’s *t* test against matched in trans controls. **** *p* < 0.0001.

**Figure 2 viruses-13-01170-f002:**
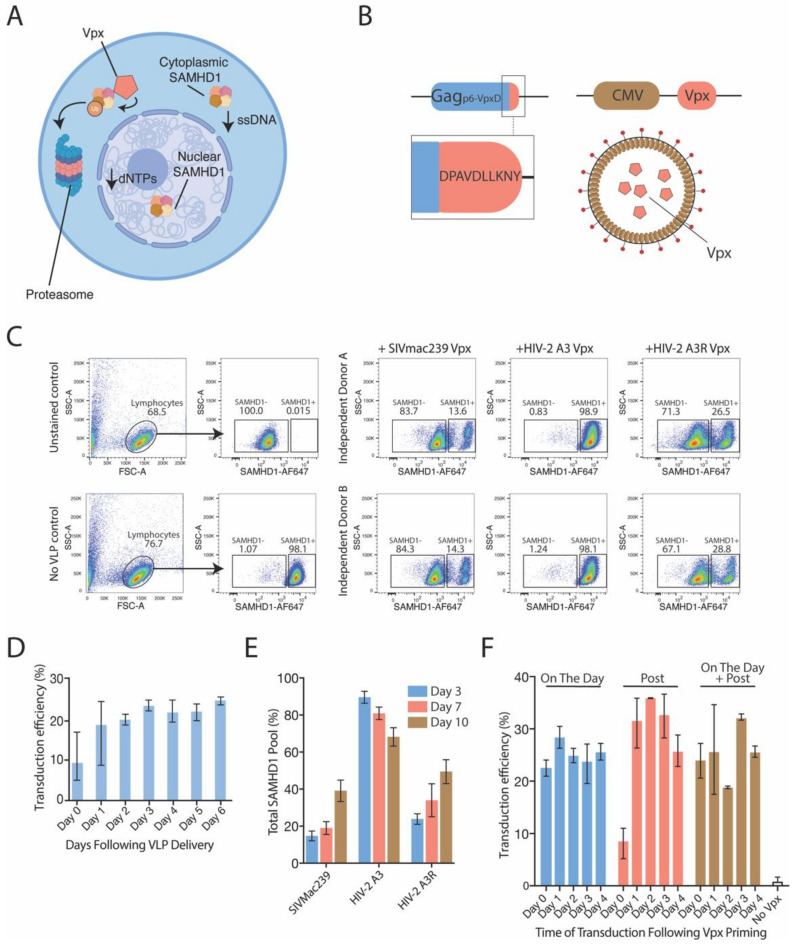
Lentiviral vectors can overcome resting CD4^+^ T cell restriction through coalescence of Vpx. (**A**) SAMHD1 is a post-entry restriction factor that limits, at two key points, the ability of lentiviral vectors to facilitate therapeutic gene transfer. Within the cytoplasm, SAMHD1 degrades dNTPs and single-stranded nucleic acids, such as vector genomes. In the nucleus, SAMHD1 further degrades dNTPs, which impairs integration. However, the viral accessory protein Vpx is able to mediate ligation of ubiquitin to SAMHD1, targeting it for degradation. This effect occurs within both the cytoplasm (shown) and the nucleus (not shown). (**B**) For the delivery of Vpx into cells, vectors could be modified within Gag to enable Vpx recognition. Next, Vpx could be expressed in trans, and coalesced into nascent particles. (**C**) Vpx derived from SIVmac239 that is delivered by unconcentrated vectors into resting CD4 T cells effectively degrades cellular SAMHD1, as revealed by flow cytometry. Vectors that coalesce A3 Vpx (a non-SAMHD1-targeting variant) do not degrade SAMHD1, but a restoration of function mutation within A3 Vpx (termed A3-Repaired; A3R) observes the reinstitution of SAMHD1 degradation. Flow cytometry was performed three days post-Vpx delivery. (**D**) Following delivery of unconcentrated vectors couriering Vpx (variant SIVmac239) into resting CD4 T cells, an increase to transduction permissibility was sustained for up to a week. (**E**) Increases to transduction permissibility is likely linked to a decrease in SAMHD1. Levels of SAMHD1 was examined temporally to ascertain whether there was timely recovery of SAMHD1 levels following delivery of Vpx. An extended period of sustained SAMHD1 depression could also sustain a window of transduction opportunity. This matched control reactions arbitrarily 100%. (**F**) Transductions efficiencies (percentage of GFP^+^ cells) were increased substantially upon Vpx delivery. Concentrated Vpx-bearing particles were delivered at three different time points; either simultaneously with a GFP-expressing vector (On the Day); 24 h following viral vector challenge (Post); or a combination of both strategies (On the Day + Post). Vpx delivered at simultaneously with vectors increased transduction potential substantially; however, cells initially challenged but that received Vpx 24 h later (Post; 0 days) still had higher transduction efficiencies compared to no-Vpx controls. Error bars on no Vpx controls show the range of transduction efficiencies observed throughout entire experiment. Unless noted, all error bars are SEM. Column bars show averages of data from at least three independent donors, with each condition performed in duplicate and averaged.

**Figure 3 viruses-13-01170-f003:**
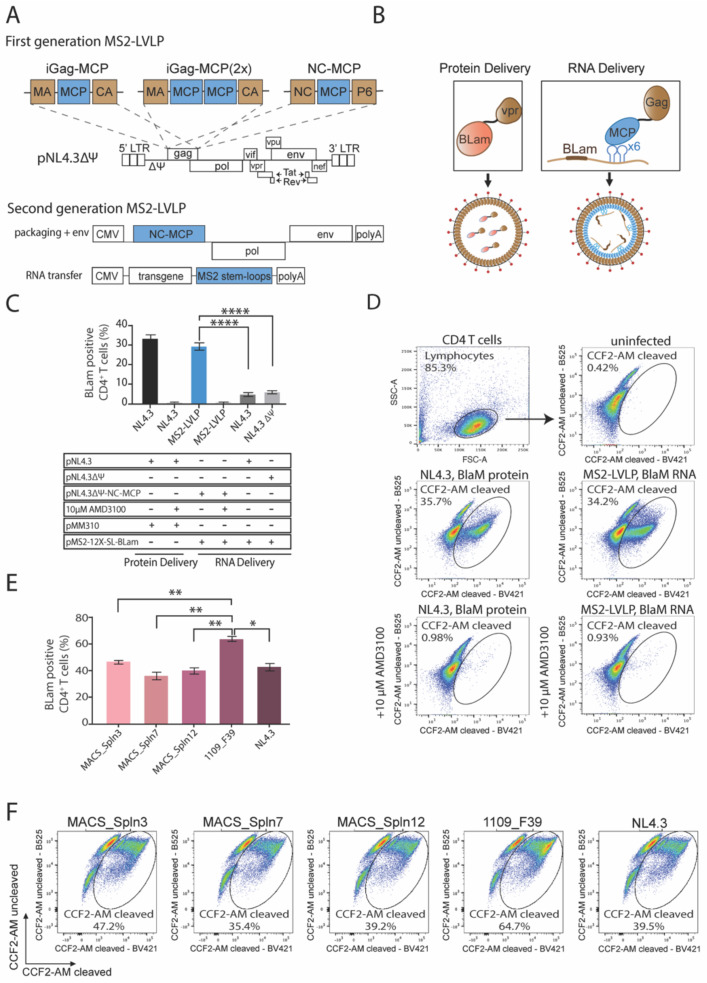
Efficient transient delivery of biologically active RNA to resting CD4 T cells using MS2-LVLPs optimized with highly fusogenic HIV envelopes. (**A**) Schematic illustration of generated first-generation and second-generation MS2-LVLPs. First-generation MS2-Gag polyprotein variants were constructed from a packaging incompetent HIV_NL4.3_ mutant (ΔΨ). The MS2CP was either incorporated in between the matrix protein (p17) and capsid protein (p24) as monomer (iGag-MCP) and head-to-tail dimer (iGag-MCP), or within the nucleocapsid protein, thereby substituting the second zinc finger (ZF2). Second-generation MS2-LVLPs with an improved safety profile are generated by co-transfection of a CMV-driven in cis packaging construct and RNA transfer construct. (**B**) Simplified representation of RNA packaging mechanism into MS2-LVLPs. MCP located on NC-MCP-HIV_NL4.3_ (ΔΨ) binds specifically to MS2 stem-loops fused to the beta-lactamase ORF on pMS2-12X-SL-BLam, leading to the incorporation of BLam RNA during particle formation. BLam protein delivery is facilitated through C-terminal fusion to vpr. (**C**) Unconcentrated first-generation MS2-LVLPs efficiently deliver BLam RNA to resting CD4 T cells. HEK 293T cells were co-transfected and LVLPs were harvested 72 h post-transfection. Resting CD4 T cells were isolated from four healthy donors and either directly transduced with unconcentrated LVLPs or incubated with 10 μM AMD3100 prior to transduction. BLam RNA delivery efficiency to resting CD4 T cells via MS2-LVLPs is significantly higher than RNA delivery efficiency of HIV_NL4.3_ and HIV_NL4.3_ (ΔΨ), highlighting the importance of the specific interaction between MS2CP and MS2-SL in RNA packaging. (**D**) Flow cytometric dot plot showing a fluorescent shift of resting CD4 T cells post-fusion assay using unconcentrated first-generation MS2-LVLPs. Live cells were gated from the Forward Scatter area (FSC-A) and the Side Scatter area (SSC-A). CD4 T cells were subsequently gated on DAPI (BV421) and FIT-C (B525). The fusion gate was drawn on the uninfected sample in a FACS plot representing cleaved CCF2-AM versus uncleaved CCF2-AM. For infected samples, cells found within the fusion gate are defined as “fused”. These cells have taken up beta-lactamase RNA or protein-carrying MS2-LVLPs, causing a shift into the ‘CCF2-AM cleaved’ channel. (**E**) A highly fusogenic HIV envelope 1109_F39 enhances the delivery of biologically active RNA to resting CD4 T cells. MS2-LVLPs were harvested from co-transfected HEK 293 T cells, and unconcentrated MS2-LVLP containing supernatant was subsequently used to transduce resting CD4 T cells. Data show the mean from three independent experiments, performed in duplicate. (**F**) Representative flow diagrams of a fusion assay performed with optimized second-generation MS2-LVLPs. Live cells were gated from the Forward Scatter area (FSC-A) and the Side Scatter area (SSC-A). CD4 T cells were subsequently gated on DAPI (BV421) and FIT-C (B525). RNA delivery data were analyzed using an ordinary unpaired t test. All data are displayed as mean ±SEM. * *p* < 0.05, ** *p* < 0.01, **** *p* < 0.0001.

**Figure 4 viruses-13-01170-f004:**
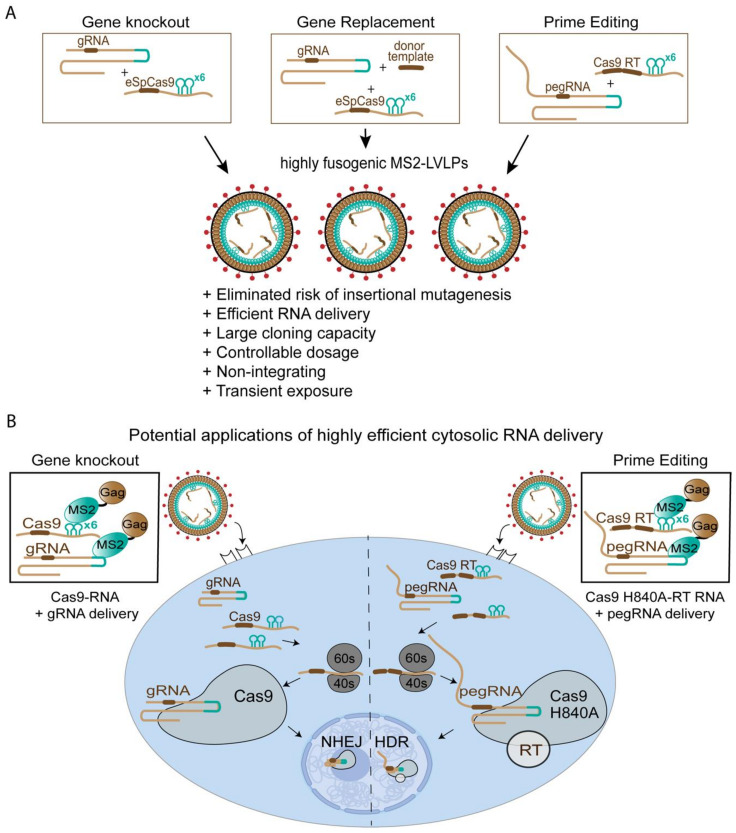
Overcoming current limitations of viral vectors in the era of genetic engineering and gene therapy through MS2-LVLP-based cytosolic RNA delivery. (**A**) Summary of advantages of transient ‘hit-and-run’ genome editing using MS2-LVLPs. (**B**) Simplified schematic of transient RNA delivery in the context of current gene editing technologies using cell-specific, non-integrating MS2-LVLPs.

**Table 1 viruses-13-01170-t001:** List of Shortlisted HIV-1 Envelopes.

Envelope Name	Tropism	CD4 Utilisation
MACS_Spln3 [20]	CXCR4/CCR5	High Affinity
MACS_Spln7 [20]	CXCR4/CCR5	Very High Affinity
MACS_Spln12 [20]	CXCR4/CCR5	High Affinity
1109_E30 [28,29]	CXCR4/CCR3	High Affinity
1109_F39 [28,29]	CXCR4/CCR3	High Affinity
1109_F42 [28,29]	CCR5	High Affinity
AD8	CCR5	High Affinity
NL4.3	CXCR4	Uncharacterised

## Data Availability

Data will be supplied following reasonable requests.

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
