# Peer review of "Modular Lentiviral Vectors for Highly Efficient Transgene Expression in Resting Immune Cells"

_viruses, 2021, doi:10.3390/v13061170_

Round 1

Reviewer 1 Report

This well-written manuscript developed novel fusogenic lentivirus-like particle (LVLP) to achieve target-specific delivery of transgene RNA or protein into cytoplasm, particularly in resting immune cells. Novel HIV Env that specifically targets resting CD4 T cells was screened and characterized previously in the same lab. Using MS2-mediated RNA binding feature, this study established MS2-LVLP to deliver either protein or RNA into the cytosol without concern about integration (oncogenic) and long-term existence (imprint). This is an excellent study with multiple innovations, careful experimental design, logistic interpretation and statistical analysis. The conclusion was supported by the strong and solid data. Many advantages of manipulating the resting (untouched) T cells over the widely used activated T cell for lentiviral transduction were nicely discussed. Excellent logistic justification for the experimental design and studies were presented. The important application of this novel technology will be the delivery of CRISRP editor or any other protein as well as mRNA vaccine.

There are minor suggestions.

In the method, a section of Statistical Analysis should be added.

In Fig. 1D, the Helper vector is misleading. Using transfer vector (or reporter vector) might be more accurate.

The MS2 immunogenicity may reduce the LVLP transduction in vivo. Also, the selected Env may encounter the preexisting antibody in HIV patients. These limits may need discussed.

Despite of beautiful English writing, a careful proofreading is needed to correct some errors. For examples, line 269, “of in” should be “in”; line 316 “of” should be “or”; line 391, “increased” should be “increase”, and many others.

Reviewer 2 Report

In this manuscript, the authors show that through modification of the lentiviral vector, increased transduction efficiency of naïve CD4 T-cells can be achieved. Utilizing alternative pseudotypes and SAMHD1 knockdown via Vpx are shown to enhance transduction of this cell type. The authors also show that MS2-lentivirus like vectors can be used to transduce naïve CD4 T-cells with high efficiency, demonstrating (1) a significant restriction at post-entry levels of transduction and (2) a useful technique that could potentially be used in CRISPR or other transient expression applications. Although this manuscript offers many promising data on achieving high transduction in a notoriously difficult cell type to transduce, many of the experiments and supporting figures are lacking crucial information in support of the conclusions the authors posit. My recommendation is to accept this article for publication only with the following major concerns and minor concerns being addressed:

Major concerns:

  • The manuscript would benefit from further editing. It is unnecessarily wordy and lengthy. There are multiple typographical errors and poorly worded sentences and sentence fragments.
  • Overall, the fonts on the figures are too small to read, causing trouble with interpreting the data. This is especially true for all flow cytometry figures and percentages (Figures 2C, 3D, and 3F). Figures and fonts in 1B, 2B, 3A, 3B, and 4B are all too small to accurately tell what is being described. Font of graphs 1D (the significance values), 1C, 3C, and 3E are too small to read.
  • The method of resting CD4 T-cell isolation is not described in the materials and methods section, and it is not shown that the isolated population the authors use are in fact resting CD4 T-cells. Flow cytometry showing markers of resting CD4 T-cells should be used in experiments in figures 2 and 3 should be included to show this is a pure population. The flow cytometry gating strategy for determining the cells used in the flow cytometry experiments of figures 2 and 3 should be included.
  • The findings in figure 2, in which the SAMHD1 positive cells are decreased upon Vpx delivery, could be an artifact of unintentional isolation of activated T cells or activation of the resting T-cell population. This could also be true for the increased transduction efficiency in figure 2F and figures 3D and 3F.
  • It is suggested that an innovative feature of the study is screening envelope pseudotypes (“bioprospecting”). Yet in figure 1 and in the methods, there is zero mention of the envelopes used in the screen for glycoproteins that show increased transduction of resting CD4 T-cells, the refined list, or the final few envelopes used in the experiment of figure 3. Without this information, the data cannot be evaluated properly, and figure 1 adds no useful information to the manuscript. This could be presented in a table. Much of the legend of Fig 1 is redundant with the text in the main manuscript.
  • In the graph of figure 1D, transduction is shown as fold change, but it is not explicit regarding what is used as the control value to create the fold change. Is it the envelope 1 in trans versus envelope 1 in cis production, etc.? This should be explicitly stated.
  • It is not described how any of vector titers are calculated, or if they are even measured. This is absolutely critical to interpreting the results. There is no mention of the specific envelopes that are used in any of the experiments, and multiple envelopes are being used in comparison of transduction efficiency. This could easily be due to differences in the amount of vector produced. For example, the increased percent of BlaM positive cells with 1109_F39 in experiment 3E could be because it simply produced more vector than the others.
  • It is mentioned in lines 213-214: “shortlisted pseudotypes were substantially more capable of entering resting T cells compared to incumbent pseudotypes” but that data is missing from the manuscript. This should absolutely be included, as it shows the ability of the unnamed envelopes to transduce this cell population as compared to classically used pseudotypes. How well do these shortlisted pseudotypes transduce resting T-cells? Is it just a small increase or a large fold-change? This is needed to appreciate the data in figure 1. If envelope pseudotyping is a focus of the paper, please tell the reader what envelopes were screened.
  • It is unclear how the Vpx is delivered to resting CD4 T-cells in figure 2C. Is this via lentiviral transduction? If so, how are “less than one third of total cells” transduced, while upwards of 80% of cells are not expressing SAMHD1? Are these data gated for transduced cells only? Is cell viability (i.e., LIVE/DEAD stain) determined? It is possible that the cells expressing SAMHD1 are merely dying off. This could be why there is not a marked increase when SAMHD1 expression is inhibited in the later experiment.
  • It is unclear if experiments in figures 2 and 3 are titer matching the vectors being used. The MOI needs to be matched to interpret any of these data.
  • It is unclear in figure 2F what “Vpx particles” refers to. Are these cells being treated with VLPs containing Vpx, or are these lentiviral vectors produced with Vpx plasmid?
  • Figures 3D and 3F are not mentioned in the text of the results. This should be remedied.
  • Lines 268-269: “Maximal transduction rates were often less than one third of total cells.” Why are these data excluded from the manuscript? This is important information that would provide context for the transduction efficiencies of this and other experiments.
  • The discussion seems to transition into a review, and Fig 4 seems completely superfluous to the content of this manuscript. There is no formal demonstration that the vectors reported here will work for gene editing, and without such data, Fig 4 seems unnecessary.
  • I think that the authors overstate the current concerns regarding the safety of lentiviral vectors. There continue to be clinical trial results that support the safety of current generation lenti vectors (see Kohn et al, NEJM, May 11, 2021).

Minor concerns:

  • Line 15-16 has an incomplete sentence. The period after 80% can be changed to a comma to remedy this.
  • Paragraph 1 of the introduction is too elementary when describing resting T-cells and activated T cells. Readers of this journal should already have a basic knowledge of these immunological pathways.
  • In line 144 of the methods of the fusion assay, the vectors used are described as VLPs. This implies that only protein is being delivered and not genetic material. This term should be changed.
  • Figure 1D shows Tat, Gag-pol, Env, and helper plasmids. The term helper plasmid makes the plasmid seems as if it is an accessory or unnecessary plasmid. The term shuttle or transfer plasmid could be used to better describe this.
  • Line 57: T cell should be changed to T cells.
  • Line 163: Delete period between lentiviral and vectors
  • Line 165: Delete period after resting.
  • Line 171: Show should be changed to shown.
  • In the figure 1 legend, it seems as though (c) and (d) are switched.
  • Line 229: Envelope should be envelope.
  • Lines 231-237 mentions that use of a plasmid expressing Gag and Env and a separate plasmid expressing pol as a Gag-pol fusion protein could also produce increased vector compared to the current 4 plasmid system. This is not tested in the results, and as it is only conjecture, should be moved to the discussion section not the results section.
  • Line 263 describes the reduction of SAMHD1 positive cells as a “near complete removal of SAMDH1” while there is still a good-sized population expressing the protein (13-28%). A more accurate descriptive term is needed here.
  • Line 275: Give should be given.
  • Figure 2C should have +A3 Vpx above column 4 of the flow cytometry data.
  • The time point of the flow cytometry of figure 2C is not stated.
  • It would be nice to see the raw data values for figures 1D and 2D in addition to fold change.
  • Line 391: Increased should be increase.
  • Line 482: Adenoviral should be adenoviral.

Reviewer 3 Report

Abstract needs some editing for more clarity, define SAMHD1. Also, the Abstract does not reflect all the findings and conclusions of the paper. Mention the broader impact of the findings, i.e. represents a strategy that can be applied to resting immune cells in general.

Line 40: rewrite- drop over time the longer cells are proliferated in culture

Materials and methods- add restriction enzyme manufacturers.

Sizes (kb) of the new final vectors generated in 2.3 should be added to the methods.

Excellent comments (line 233 and on) on the safety of a cis simplified 2-plasmid system for self replicating virions. Perhaps expand with further comments or references using 2-plasmid systems as to augment confidence this is safe.

Line 275-comment on whether there are other proteins related to SAMHD1 that may be compensating for its activity in Lymphoid cells

Line 395- expand some more on the identity or examples of the elements/proteins that can be useful for potential T cell tropic envelopes.
